# A Metric for Questions and Discussions Identifying Concerns in Software Reviews

**Michiyo Wakimoto** *[ID] **and Shuji Morisaki**

Graduate School of Informatics, Nagoya University, Nagoya 464-8601, Japan
* Correspondence: wakimoto.michiyo@g.mbox.nagoya-u.ac.jp

**Abstract:** Previous studies reported that reviewers ask questions and engage in discussions during software reviews and that the concerns identified by the questions and discussions help detect defects. Although such concerns about potential defects lead to finding defects, review metrics such as the number of defects detected do not always reflect the questions and discussions because concerns which are not applicable to the review material are excluded from the number of defects. This paper proposes a metric, the number of questions and discussions, which identifies concerns in reviews. First, we defined an effective question, which identifies concerns. Then, we defined detailed review processes (identifying, sharing, and recording processes), which capture how concerns identified by effective questions are shared and defects are documented. We conducted a case study with 25 projects in industry to investigate the impact of the number of effective questions, which identified concerns on the number of detected defects in subsequent testing. The results of a multiple regression analysis show that the number of effective questions predicts the number of defects in subsequent testing at the significance level of 0.05.

**Keywords:** effective questions; concerns; software reviews; software metrics; software quality



## 1. Introduction

Software review is a static analysis technique aimed at the early detection of defects [1–4]. Software review is also one of the most effective evaluation techniques of quality assurance [5–8]. Reviewers manually check materials (documents and source code) in a review to ensure that no defects remain [9]. Specifically, reviewers point out potential defects, and then the authors and reviewers verify that they are true defects, which require action, including correction. Typically, defects overlooked in reviews are detected and corrected in subsequent development activities, including testing.

Reviewers not only detect defects but also ensure that the review material is free of concerns about potential defects by asking questions and engaging in discussions, because the concerns may cause defects [10,11]. A study which analyzed utterances during reviews [12] reported that 60–70% of the conversations consisted of "informing" and "clarification". Other studies [13,14] reported that reviewers spend 38% of the review time verifying, justifying, or rejecting potential defects (concerns). A study on code review effectiveness [15] reported that code review comments included questions, and these questions helped reviewers detect defects. In the case where a concern identified by a question is applicable to the review material during the subsequent discussion, the concern and applicable locations are specified as a defect. On the other hand, in the case where a concern is not applicable to the review material, it is discarded or recorded as a false-positive defect. For example, in a code review, a concern may be identified by the question, "Is it intentional that one of the parameters passed to the function is not used?". Then, the subsequent answers and discussions enable the reviewers and authors to find that the source code statements using the parameter passed to the function are omitted. In this case, the concern "the implementation using the parameter passed to the function may be

omitted" identified by the question and the discussion reveals a defect: "the implementation using the parameter passed to the function is omitted". On the other hand, if the parameter passed to the function is designed for compatibility with older versions and is not used intentionally, the concern is not applicable and will not identify a defect. Although this kind of question and discussion may lead to defect detections, its effectiveness and the detailed process have yet to be investigated.

Concerns identified by questions and discussions cannot be directly extracted from defects in a defect list after reviews because the defects include defects directly detected by reviewers and defects found by examining concerns. Furthermore, some concerns are discarded or recorded as false-positive defects if they are not applicable to the review materials. Although some studies have used objective indicators such as the number of detected defects to assess whether reviews are performed properly [16–18], such metrics only include the number of defects directly detected by reviewers and defects found by concerns, which are applicable to the review material.

The number of questions identifying concerns can be an indicator for effective reviews. Some studies have demonstrated that the number of questions identifying concerns is an indicator of an effective review. One study evaluated the percentage of interrogative sentences in each review comment as a metric for code review quality [19]. Another study defined a new metric, Issue Density, to estimate the code review quality [15]. However, neither study evaluated the relationship between the quality of review and the quality in subsequent testing.

This paper proposes a metric, the number of effective questions which identify concerns in reviews. First, we defined effective questions and the processes by which effective questions are recognized and recorded as defects as well as the categories for true and false-positive defects. Then, we surveyed previous studies according to the defined process and defect categories to investigate whether defects are distinguished defects directly detected from those found by concerns according to the defined processes and categories. Furthermore, we implemented a case study, which involved 25 projects in industry, to investigate the effectiveness of effective questions in reviews. The metrics in the case study include the number of effective questions that identified concerns, number of defects detected, and number of defects detected in subsequent testing. We performed multiple regression analysis for these metrics to evaluate the effectiveness of questions and discussions in software reviews. The research questions are formulated as the following.

RQ: Does the number of effective questions in a review affect the quality of subsequent testing?

This paper is structured as follows. The software reviews are described in Section 2. Section 3 defines the effective questions and discussions in software reviews and processes for identifying, sharing, and recording in software reviews. Section 4 conducts a case study to investigate the impact of the number of effective questions on the number of defects detected in subsequent testing. Section 5 discusses the results, and Section 6 summarizes this paper.

## 2. Software Reviews

Guided reviews are one approach to detecting defects in software reviews. Guided reviews help reviewers comprehensively detect severe defects, including omissions or ambiguities, by providing detailed instructions, procedures, and hints [20]. Many studies have reported on the effectiveness of guided reviews [4,21–29]. Typical techniques of guided reviews are checklist-based reading (CBR) [1], perspective-based reading (PBR) [25], defect-based reading (DBR) [24], usage-based reading (UBR) [26], and traceability-based reading [30]. CBR is a reading technique in which reviewers use a list of questions to help them understand what defects to examine [27]. PBR [21,31,32] is a scenario-based reading (SBR) [24] that defines the perspectives of the stakeholders and assigns the perspectives to reviewers. DBR is an SBR that focuses on detecting specific types of defects [24,27]. UBR prioritizes the use cases and detects the most critical defects in the target materials along with the prioritized use cases [27,30]. While reviewers in these reading techniques ask

effective questions and engage in the following discussions, no studies referred to effective questions and discussions. Investigating the effective questions and discussions leads to higher predictions of review quality.

## 3. Effective Questions in Reviews

### 3.1. Definition

We define effective questions to distinguish between questions that identify concerns about potential defects from those that clarify and understand the review material, because questions and subsequent discussions in reviews cover diverse topics such as exchanging opinions on defects, evaluating the value, clarifying solutions, and rejecting hypotheses [13,14]. Namely, a set of questions ($U$) consists of a set of effective questions ($D_{iq}$) and a set of non-effective questions ($N$). We assume that a defect can be specified with a concern and its locations in the review material. The discussions following the effective questions ($D_{iq}$) verify the concern and determine the locations applicable to the concern. Thus, if a concern identified by an effective question is judged to apply to the review material through discussions, the locations of concern can be determined. On the other hand, if a concern is not judged to apply to the review material through discussions, no location is determined. Thus, the concern does not lead to specifying (finding) a defect. Namely, a set of effective questions ($D_{iq}$) consists of a set of effective questions identifying concerns with applicable locations ($D_{iql}$) and a set of effective questions identifying concerns without applicable locations ($D_{iqn}$). Figure 1 shows the flowchart for categorizing effective questions and determining true or false-positive defects identified and specified by effective questions. The first and second branches categorize effective questions. As indicated by the second branch in Figure 1, if reviewers do not attempt to find the applicable locations for the concern identified by the question, then it is considered to be a non-effective question. An example of a non-effective question without a concern is "What time does this review meeting end?". An example of a non-effective question for the reviewer's self-understanding is "Which chapter defines the glossary?".

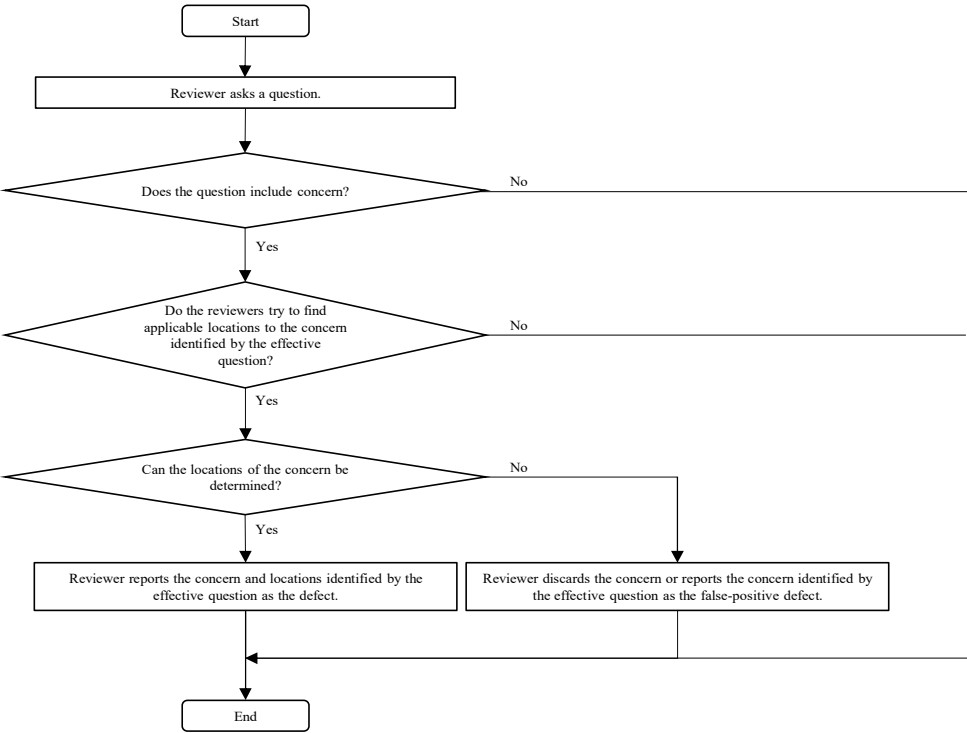

**Figure 1.** Flowchart to categorize effective questions and distinguish between true and false-positive defects identified and specified by effective questions.

Reviewers ask an effective question when they cannot specify locations for concerns or when they are unable to expend effort to check and find locations of concerns. Figure 2 shows an example of an effective question. An example is the question ($\in U$), "Does the interrupt program change the value of the global variable x? If yes, the assigned value and reference value are not consistent". This identifies a concern that the global variable x can be overwritten by the interrupt program. If the interrupt program, which changes the global variable x, can be executed during the assignment and reference, the concern applies to the review material (source code A in Figure 2). The locations are where the interrupt program changes the value of the global variable x or the omitted place (description), disabling the interrupt. Then, the defect ($\in D_{st}$) "The value of the global variable x may not be consistent because the interrupt program can change the value, and disabling the interrupt programs is omitted". is detected by the effective question, identifying the concern applicable to locations ($\in D_{iql}$). In the case where a concern identified by an effective question applies to the review material, the defect is recorded as a true defect ($\in D_{rt}$). If a concern does not apply to the review material (source code B in Figure 2), the concern is discarded or recorded as a false-positive defect ($\in D_{rf}$) detected by the effective question identifying the concern without applicable locations ($\in D_{iqn}$), depending on the recording policy.

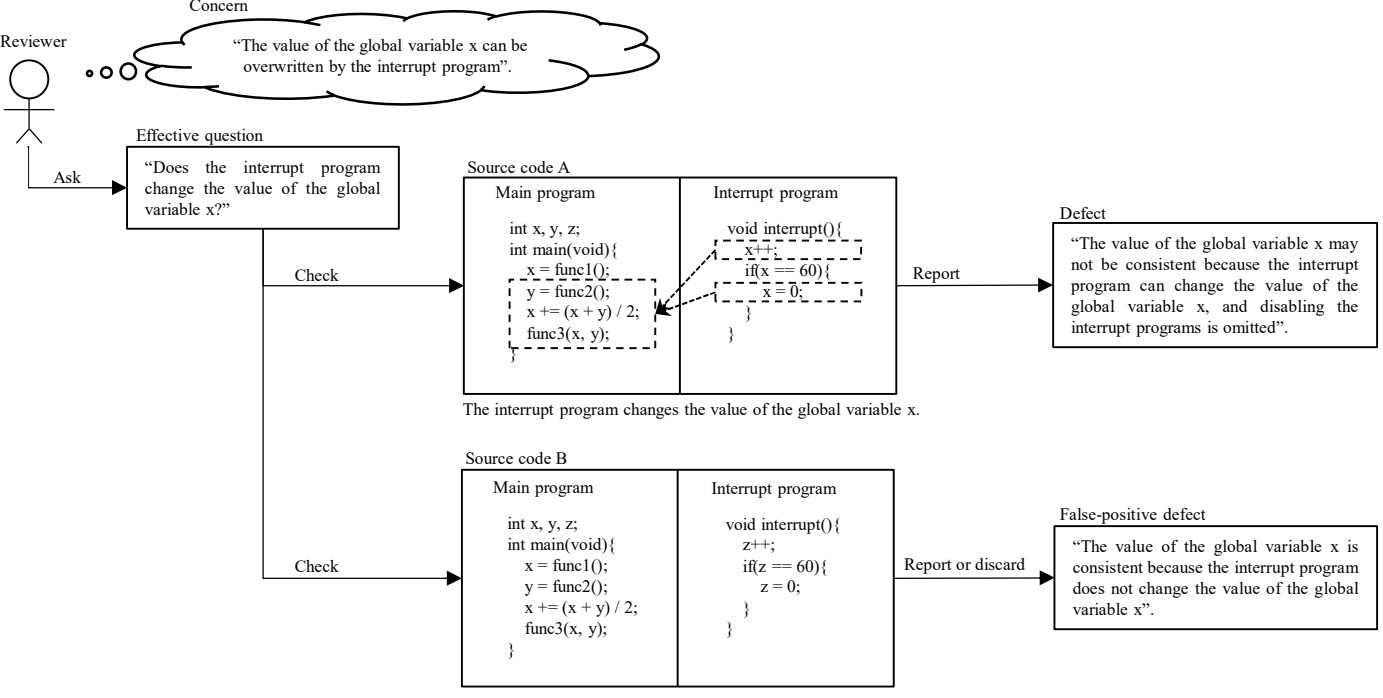

**Figure 2.** An example of an effective question, which shows true or false-positive defects from an effective question.

The number of effective questions identifying concerns can be an indicator for effective reviews. Furthermore, reviewers are expected to directly detect defects, and they then have confidence to ask effective questions. The proportion of the number of effective questions to the number of directly detected defects should be measured because available time and effort for the reviews are limited. If the proportion of the number of detected defects to the number of effective questions is larger, time and effort for asking questions are likely to be limited.

*3.2. Defect Category and Effective Questions in Review Process*

Reviews can be categorized as synchronous, such as a face-to-face meeting, or asynchronous, such as sending and receiving defect descriptions via a review support tool [33–35]. In synchronous reviews, reviewers share potential defects and ask effective questions in a review meeting. In asynchronous reviews, reviewers share potential defects and ask effective questions using review support tools. Although reviewers present potential defects and ask effective questions in both synchronous and asynchronous reviews, their processes differ.

In synchronous reviews, reviewers present potential defects and ask effective questions during a review. According to Fagan, a review consists of overview, preparation, review, rework, and follow-up processes [1]. In the review process, a reviewer presents potential defects and asks effective questions. Then, the authors respond. If necessary, the potential defects and concerns identified by the effective questions are further discussed [1,36].

In asynchronous reviews (non-meeting-based approaches [35]), a reviewer denotes potential defects and effective questions, which are sent to the authors and other reviewers via a review support tool. After the authors answer the effective questions, the authors and reviewers discuss the concerns identified by the effective questions and answers using the tool. In asynchronous patch reviews, a fix proposal (a code patch) may be sent with a potential defect [37,38].

Both synchronous and asynchronous reviews include the following identifying, sharing, and recording processes.

- Identifying

A reviewer checks the material and identifies potential defects. It is assumed that the reviewer thinks that the potential defects are true defects, as the reviewer does not want to share false-positive defects in reviews. If the reviewer has a concern, they prepare effective questions, which will be asked in the sharing process. In asynchronous reviews, the reviewer inputs the potential defects and effective questions into the review support tool.

- Sharing

Each potential defect identified by the reviewers is shared, and whether it is a true or false-positive defect is evaluated. The authors and other reviewers answer the effective questions and discuss the identified concerns to find applicable locations and ensure that no defect remains. Each potential defect or concern identified by effective questions is subsequently categorized as either a true defect or a false-positive defect.

- Recording

In synchronous reviews, the true defects judged in the sharing process are recorded. In some reviews, defects judged to be false positives in the sharing process are recorded, whereas in other reviews, they are discarded. In asynchronous reviews, potential defects and effective questions are already recorded in the identification process. Hence, potential defects and effective questions are categorized into true or false-positive defects. In addition, the defect descriptions may be updated, depending on the discussions in the sharing process.

Figure 3 overviews the process to categorize potential defects and effective questions in the sharing process and how true and false-positive defects are recorded in the recording process. For synchronous reviews, reviewers identify potential defects ($D_{id}$) and effective questions ($D_{iq}$) in the identification process. In the sharing process, the reviewers present $D_{id}$ and ask $D_{iq}$, and then the authors and the other reviewers examine defects ($D_{id}$) and concerns identified the by $D_{iq}$. Finally, based on their discussion, the defects and concerns are categorized into true defects ($D_{st}$) and false-positive defects ($D_{sf}$). In the sharing process, new potential defects and effective questions may be found. In this case, they are added to $D_{id}$ and $D_{iq}$. In the recording process, each defect in $D_{st}$ is recorded as true defects ($D_{rt}$). Depending on the recording policy, some defects in false-positive defects ($D_{sf}$) are

recorded as false-positive defects ($D_{rf}$). After the recording process, defects in true defects ($D_{rt}$) are corrected.

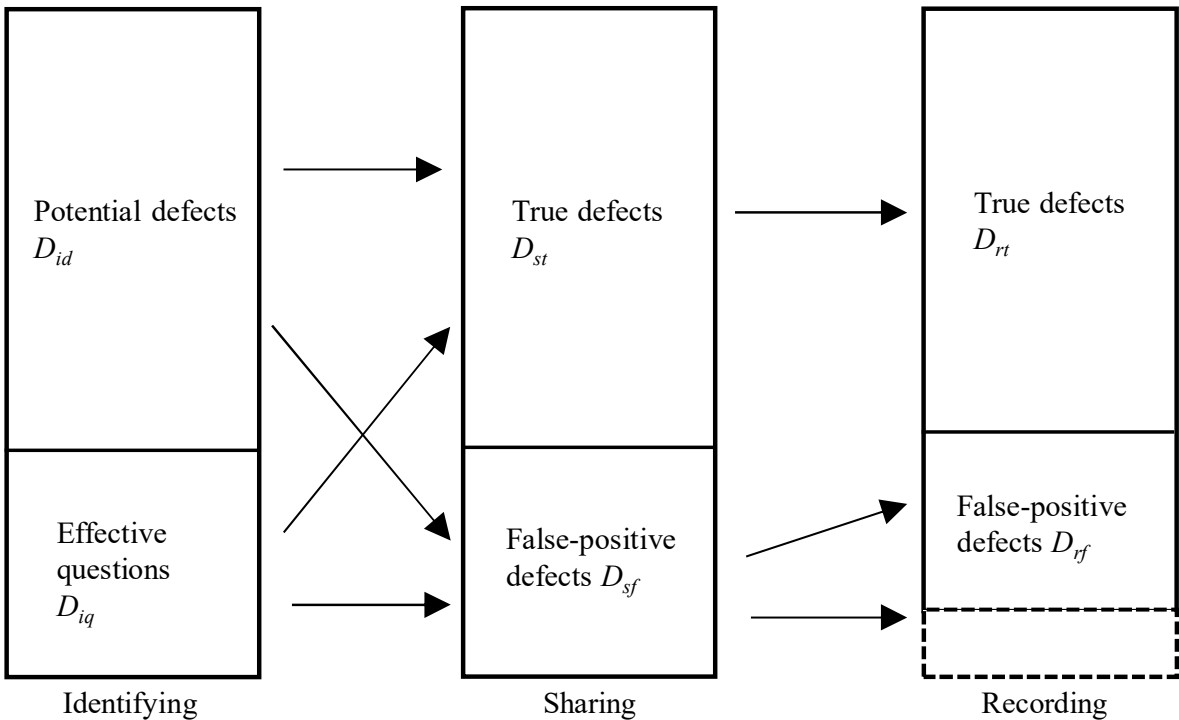

**Figure 3.** Defect categories in identifying, sharing, and recording process.

For asynchronous reviews, reviewers identify potential defects ($D_{id}$) and effective questions ($D_{iq}$). Then, they input $D_{id}$ and $D_{iq}$ into a review support tool. In the sharing process, the reviewers send $D_{id}$ and $D_{iq}$ to the authors and other reviewers. After the authors and other reviewers understand $D_{id}$ and $D_{iq}$, they answer the effective questions ($D_{iq}$) and discuss the concerns identified by $D_{iq}$. In addition, they examine and judge whether the potential defects ($D_{id}$) are true defects.

Finally, the authors and the reviewers categorize $D_{id}$ and $D_{iq}$ into true defects ($D_{st}$) and false-positive defects ($D_{sf}$) based on the discussions. Effective questions ($D_{iq}$) are categorized into effective questions identifying concerns with applicable locations ($D_{iql}$) and effective questions identifying concerns without applicable locations ($D_{iqn}$) depending on whether the concerns are applicable or not. In the recording process, true defects ($D_{st}$) are labeled or categorized as true defects ($D_{rt}$). False-positive defects ($D_{sf}$) are labeled or categorized as false-positive defects $D_{rf}$. Some of the defects in the false-positive defects ($D_{sf}$) may be discarded in the recording process. After the recording process, true defects ($D_{rt}$) are corrected. In the case where a code patch is attached to the true defects ($D_{rt}$), the patches are merged. Figure 4 shows an example for the process and categories. In Figure 4, if the locations of concern are found through discussion, the author (reviewee) who knows the design intention answers, "The implementation using the parameter passed to the function is omitted". On the other hand, if the locations of concern are not found through discussion, the author (reviewee) answers, "The parameter is designed for compatibility with older versions and is not used in the function intentionally".

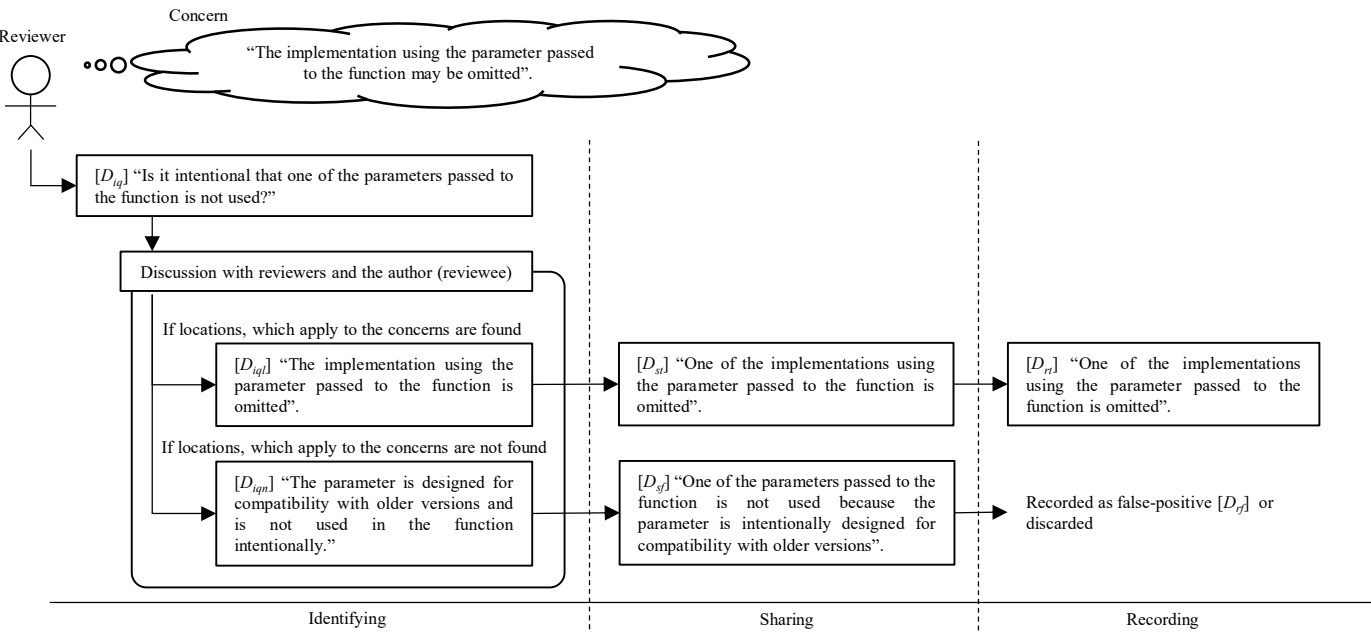

**Figure 4.** An example for the process and categories, which shows the categorization of effective questions according to the flowchart in Figure 1 and the process in Figure 3.

### 3.3. Literature Review

We conducted a literature review to identify articles that describe concerns raised by effective questions, subsequent discussions, and categorization of true and false-positive defects according to the applicable concerns. Many studies categorized true defects [39–41], but few studies categorized false-positive defects. Previous studies referring to reviews in which false-positive defects were detected [9,36,39,42] did not refer to the categories or details of false-positive defects. Articles [9,43] excluded false-positive defects prior to defect analysis. Moreover, one study described that analysis and retrospect of reviews used information of true defects as inputs [22]. Only the article [44] referred to the use of a question list. However, it did not describe the concerns identified by the questions.

We investigated which processes and categories defined in Sections 3.1 and 3.2 are referred to when true defects and false-positive defects are judged and recorded. Table 1 shows the result. True defects and false-positive defects were judged and recorded in different processes. Three articles [44–46] referred to defects in $D_{id} \cap D_{st}$. These articles described that the participants of the reviews discussed whether the presented defects were true defects or not, and they categorized defects as true defects. One article [47] referred to defects categorized as $D_{id} \cap D_{sf}$. It described that the participants in the reviews discussed whether the presented defects were false-positive or not prior to deciding they were false-positive defects. One article [48] referred to defects categorized as $D_{id} \cap (D_{st} \cup D_{sf})$. The participants of the reviews discussed whether the presented defects were false-positive defects or not, and they found both true and false-positive defects. One article [45] referred to defects categorized as $D_{st}$ and $D_{sf}$. Another article [44] referred to defects categorized as $D_{st} \cap D_{rf}$. One article [39] referred to defects categorized as $D_{rf}$.

**Table 1.** Defect processes in previous studies.

| Categories | Text Referring the Defect Categories in the Previous Studies |
|---|---|
| $D_{id} \cap D_{st}$ | To support decision making, discussants can also vote by rating any potential defect as true defect [44]. Collated defects: the number of defects merged from individual findings to be discussed during the meeting. True defects: the number of defects for which consensus was reached during the meeting in considering them as true defects [45]. We used the information from the repair form and interviews with the author to classify each issue as a true defect (if the author was required to make an execution affecting change to resolve it) [46]. |
| $D_{id} \cap D_{sf}$ | False positives are items reported by subjects as defects, when in fact no defect exists [47]. |
| $D_{id} \cap (D_{st} \cup D_{sf})$ | In addition to the instructions from the preparation phase, the instructions in the meeting phase were: use the individual inspection record and decide which are faults and which are false positives [48]. |
| $D_{st}$ | True defects: the number of defects for which consensus was reached during the meeting in considering them as true defects [45]. |
| $D_{sf}$ | Removed false positives: the number of defects for which consensus was reached during the meeting in considering them as not true defects, thus as false positives [45]. |
| $D_{st} \cap D_{rf}$ | In the Discrimination stage, discussion takes place asynchronously as in a discussion forum. When a consensus has been reached, the moderator can mark potential defects as false positives, thus removing them from the list that will go to the author for rework (potential defects marked as false positives appear strikethrough in Figure 4) [44]. |
| $D_{rf}$ | False positives were issues that were identified in the meeting but that were discovered not to be defects either during the meeting or after. The decision whether a defect was a false positive was done by the code review team [39]. |

Table 2 summarizes the result of the literature review. Table 2 shows the definitions of false-positive defects, percentages of false-positive defects, and definitions of true defects for each article. We searched the percentage of false-positive defects because the percentage of false-positive defects equals the maximum percentages of effective questions, which did not apply to the review material when all of the effective questions were categorized into false-positive defects.

We investigated the definitions of false-positive defects to survey the categories for false-positive defects. Table 2 shows the definitions of true and false-positive defects. No article categorized false-positive defects into incorrectly detected defects and concerns that are not applicable to the review material. Eighteen articles described the definitions of both true and false-positive defects. No article described effective questions. One article [44] presented the format of a question list in a software review, but it did not refer to concerns identified by the questions in the question list.

Table 2 shows that not only the definitions of false positives but also those of true defects were inconsistent among the articles. Table 2 also shows the percentages of false-positive defects to the sum of true and false-positive defects. The percentages varied from 20% to 80%. More than half of the articles did not indicate the percentages or refer to the false-positive defects. Ten articles referred to categorizing detected defects into true and false-positive defects but did not report the percentages of the false-positive defects. Eight articles reported the percentages of detected false-positive defects.

The results of the literature review showed that no article referred to questions and concerns that were categorized into true or false-positive defects. Additionally, no literature categorized true defects into defects directly detected and shared by a reviewer or those found by concerns identified by effective questions and subsequent discussions. Therefore, we investigated whether the number of questions identifying concerns leads to an indicator for effective reviews and helps a project manager identify an insufficient review.

**Table 2.** Definitions and percentages of false-positive defects.

| Article | Definitions of False-Positive Defects | Percentages of False-Positive Defects | Definitions of True Defects |
|---|---|---|---|
| [9] | False positives (issues raised as defects that are not actual defects) False positives, the number of invalid defects recorded by the group | 22% | Defects, the total number of distinct, valid defects detected by a group |
| [36] | False positives (issues raised as defects that are not actual defects) | 22% | Actual defects |
| [39] | False positives were issues that were identified in the meeting but that were discovered not to be defects either during the meeting or after | 22% | If the code review team finds an issue and agrees that it is a deviation from quality, the issue is counted as a defect |
| [42] | False positives (no real usability problems) | 43.10% | Real usability problem |
| [43] | False positives (reported defects that were not considered to be actual defects) | - | Actual defects |
| [44] | False positives (non-true defects) False positives (defects erroneously reported as such by inspectors) | 46% | True defects |
| [45] | For which consensus was reached during the meeting in considering them as not true defects, thus as false positives | - | True defects: the number of defects for which consensus was reached during the meeting in considering them as true defects |
| [46] | False positive (any issue which required no action) | 20% | True defect (if the author was required to make an execution affecting change to resolve it), soft maintenance issue (any other issue which the author fixed) |
| [47] | False positives are items reported by subjects as defects, when in fact no defect exists | - | Defects |
| [49] | A false positive is a description which is not a true defect, i.e., does not require rework | - | A true defect is a description of a positively identified defect which requires rework; it causes the program to fail, and violates the given specifications and design |
| [50] | False positives (erroneously identified defects) False positives are the non-true defects—defects that require no repair | 42.62% | True defects |
| [51] | It classifies too many consistent designs as inconsistent (false positives) | - | True positive |
| [52] | False positive (FA)—defects that do not exist but were wrongly identified | - | True defects (TR)—defects that actually exist and have been successfully detected |
| [53] | False defect estimations, known as false positive | - | The number of true defect estimations, known as true positive |
| [54] | False positive rate: the percentage of issues reported by an inspector that turn out not to represent real quality problems in the artifact | 80% | Defect detection rate: the percentage of known defects in a given software artifact that are found during the inspection |
| [55] | False positives (not identified from preparation) | - | True defects Net defects |
| [56] | A false positive—an obviously wrong statement of the document. | - | True defect |
| [57] | False positive—items pointed by the subjects that do not correspond to a defect of the RD RD: the Requirements Document | - | Defects—items that really are defects of the RD RD: the Requirements Document |

## 4. Case Study

### 4.1. Goal

This evaluation investigated whether the number of effective questions in reviews predicts software quality. Specifically, the metric defect detection rate in testing ($Q$) was used as the quality of the software, where $Q$ = [number of defects detected in testing]/[lines of source code]. The evaluation examined whether the number of effective questions predicted the defect detection rate in testing $Q$ by performing multiple regression analysis because multiple parameters may affect $Q$. The independent variables of the multiple regression analysis include the number of effective questions in the reviews. This evaluation assumes that effective reviews decrease the number of defects detected in testing because effective design and code reviews reduce defects overlooked in the reviews. Consequently, defects detected in subsequent testing are reduced.

### 4.2. Projects

The data for the evaluation were collected from a Japanese software development Company *S*. The standard software development process in Company *S* is based on the waterfall model and follows the process areas Organizational Process Definition (OPD) and Integrated Project Management (IPM) defined in CMMI-DEV V.1.3. The standard process also defines software measurements and metrics. In each software development project in Company *S*, the standard development processes require that detected defects and review logs including review comments in reviews should be recorded in a defect list and that the detected defects in testing should be recorded, too.

The standard software development process of Company *S* requires that each project performs design and source code reviews. The reviews are performed in a synchronous (face-to-face meeting) or asynchronous (adding detected defects to defect lists on a defect tracking server) manner. The standard process of Company *S* also requires that each reviewer complete review training and have detailed knowledge on the product domain to participate in a review.

The evaluation used metrics collected in 25 projects of Company *S*. First, we selected 33 completed projects between April 2010 and March 2016 in Company *S*. Second, for each of the 33 projects, we checked that the metrics did not have missing values for review metrics, review logs, and defect metrics in testing. Eight projects were excluded due to the missing values. Finally, we measured the number of effective questions categorized as false-positive defects from the review logs of the remaining 25 projects. The reviewers of the case study categorized the effective questions into true defects or false-positive defects. If the reviewers categorized the effective questions into true defects, they were recorded as true defects in the defect list. A quality assurance team in company *S* verified the categorizations.

The 25 projects were for the development of embedded systems software, including safety-critical systems software, specifically, communication control systems software, engine control systems software, and browsing systems software. The development types were new development from scratch, enhancement of the same product, and reuse from another product. The number of project members varied from 3 to 20. The number of years of software development experience of the project members varied from 1 to 25 years. The lines of source code varied from 3000 to 1,100,000 lines written in C, C++, or Java.

### 4.3. Metrics and Procedure

Table 3 shows the metrics, excluding the number of effective questions categorized as false-positive defects, collected for project management defined by the standard development process. The product size (SZ) was used to assess the project progress management defined in the standard software development process. In Table 3, SZ is equal to the lines of code developed in the project without reusing code. In the development of enhanced or evolved development projects reusing an existing code base, SZ is equal to the sum of the lines of newly developed code (nLOC), lines of changed code from the code base

(cLOC), and lines of reused code (rLOC) with a coefficient. The standard development process determines the coefficient according to the project attributes, such as the product domain and development types, to assess effort consumption to the product size in the project management. The number of effective questions categorized as false-positive defects (NOQf), not the number of effective questions, was measured because the effective questions categorized as true defects and rNOD would be double-counted. The metric of NOQf was measured from the review logs. The standard development process defines that review logs should include questions, which affect the quality of the product because some of the products in Company *S* are embedded in safety-critical systems. Consequently, the review logs could be used as a part of accountability for safety, if needed.

**Table 3.** Measured metrics for project management.

| | Name | Description |
|---|---|---|
| Lines of code | New (nLOC) | Lines of code newly developed, excluding headers and comments |
| | Changed (cLOC) | Lines of code changed from the code base or reused source code, excluding headers and comments |
| | Reused (rLOC) | Lines of code reused from the code base or another product, excluding headers and comments |
| Product size (SZ) | | Product size for assessing development effort consumption in the project management defined by the standard development process. SZ = nLOC + cLOC + rLOC × coefficient (where the coefficient is determined by the project attributes) |
| Number of defects and questions in reviewed | True defects (rNOD) | Sum of the number of defects detected in software architecture design, software detailed design, and code reviews |
| | Effective questions categorized as false-positive defects (NOQf) | Sum of the number of effective questions subsequently categorized as false-positive defects detected in software architecture design, software detailed design, and code reviews |
| Number of defects detected in the test (tNOD) | | Sum of numbers of defects detected in the unit test, software integration test, and software qualification test |

Table 4 shows the derived metrics from the metrics shown in Table 3 for this evaluation. The dependent variable was $Q$, which measured the software quality in the standard software development process, because it was an indicator of the software quality in Company *S*. The independent variables included the proportion of rLOC to the total lines of code ($p_1$), and proportion of the number of true defects detected in reviews (rNOD) to SZ ($p_2$). These independent variables were used in the project management and were defined in the standard software development process. The denominator of $p_1$ was nLOC + cLOC + rLOC. The standard software development process included metric $p_1$ because it was an indicator to estimate the productivity and had a higher correlation with the number of detected defects in the past developments. The standard software development process included metric $p_2$ because it was used as an evaluation criterion to measure the effectiveness of reviews. The remaining independent variables, proportion of NOQf to SZ ($p_3$), and the proportion of NOQf to the sum of rNOD and NOQf ($p_4$) were measured. If NOQf was large (large $p_3$), true defects were likely to be overlooked in the reviews because the discussions and concerns may have missed the point. If the proportion of NOQf to the sum of rNOD and NOQf was large (large $p_4$), true defects were likely to be overlooked. The review time to detect the true defects was insufficient when the value of $p_4$ was large and the review time was a constraint. For metric $p_3$, we normalized NOQf by SZ because they largely depended on SZ as well as the independent variable $p_2$. For metric $p_4$, as described in Section 3, we normalized NOQf by the sum of rNOD and NOQf because rNOD may affect NOQf in the reviews. Specifically, the sum of rNOD and NOQf was likely to be limited due to available time and effort for the reviews.

**Table 4.** Derived metrics for the analysis.

| | Name | Description |
|---|---|---|
| $Q$ | Proportion of the number of defects detected in testing to the product size | tNOD/SZ |
| $p_1$ | Proportion of the reused lines of code to the lines of code | rLOC/(nLOC + cLOC + rLOC) |
| $p_2$ | Proportion of the number of true defects to the product size | rNOD/SZ |
| $p_3$ | Proportion of the number of effective questions categorized as false-positive defects to the product size | NOQf/SZ |
| $p_4$ | Proportion of the number of effective questions categorized as false-positive defects to the sum of the number of defects and effective questions categorized as false-positive defects | NOQf/(rNOD + NOQf) |

In the multiple regression analysis, we selected significant independent variables using the stepwise method. The evaluation investigated whether metrics of NOQf ($p_3$ and $p_4$) predicted the metric of the number of detected defects in testing ($Q$).

*4.4. Results*

Table 5 shows the distribution of the measured metrics. Table 6 shows the distribution of the dependent and independent variables. Table 7 shows the results of the multiple regression analysis. Metrics $p_1$, $p_3$, and $p_4$ contained significant coefficients. The variance inflation factor (VIF) values indicated that there was no multicollinearity among the variables. The adjusted $R^2$ of the model was 0.45 ($p = 0.0013$).

**Table 5.** Distribution of the measured metrics.

| | SZ | rNOD | NOQf | tNOD |
|---|---|---|---|---|
| max | 110,5000 | 1871 | 384 | 711 |
| min | 1330 | 32 | 0 | 3 |
| median | 144,390 | 589 | 99 | 171 |

**Table 6.** Distribution of the independent and dependent variables.

| | $Q$ | $p_1$ | $p_2$ | $p_3$ | $p_4$ |
|---|---|---|---|---|---|
| max | 6.50 | 0.97 | 46.29 | 7.90 | 0.29 |
| min | 0.54 | 0.00 | 0.55 | 0.00 | 0.00 |
| median | 2.61 | 0.75 | 12.03 | 1.42 | 0.17 |

**Table 7.** Results of the multiple regression analysis.

| | Estimate (b) | Std. Error | t Value | Pr (>\|t\|) | VIF |
|---|---|---|---|---|---|
| $p_1$ | 2.53 | 0.88 | 2.89 | 0.01 | 1.07 |
| $p_3$ | 0.42 | 0.15 | 2.91 | 0.01 | 1.38 |
| $p_4$ | −9.68 | 3.69 | −2.63 | 0.02 | 1.46 |

From the coefficients in Table 7, the model is expressed as

$$Q = 1.71 + 2.53p_1 + 0.42p_3 - 9.68p_4$$

The metrics of NOQf ($p_3$ and $p_4$) affected $Q$. The proportion of NOQf to SZ ($p_3$) increased $Q$. The proportion of NOQf to the sum of rNOD and NOQf ($p_4$) decreased $Q$. Specifically, the metric $p_3$ (ranging from 0.00 to 7.90) increased $Q$ (ranging from 0.54 to 6.50). The coefficient of $p_3$ was 0.42 ($p = 0.01$). The metric $p_4$ (ranging from 0.00 to 0.29) decreased $Q$. The coefficient of $p_4$ was −9.68 ($p = 0.02$).

## 5. Discussion

*5.1. RQ: Does the Number of Effective Questions in a Review Affect the Quality of Subsequent Testing?*

The results of the case study indicated that the answer to RQ is yes. In the case study, $p_3$ (the proportion of NOQf to SZ) positively affected $Q$ (tNOD to SZ). We did not assume that $p_4$ (the proportion of NOQf to the sum of rNOD and NOQf) negatively affected $Q$ (the proportion of the number of defects detected in subsequent testing (tNOD) to SZ). Hence, we investigated the review details. We found that the review materials contained a small number of defects. Almost all defects were detected in the reviews. The number of defects detected in subsequent testing was small. Furthermore, the reviewers took a shorter time to detect almost all the defects, indicating that reviewers had enough time to ask additional effective questions to ensure that they did not overlook the remaining defects. In the discussion with the reviewers, they indicated that the potential true defects were shared before they asked effective questions and discussed them in higher quality projects. This suggests that sharing potential true defects has a higher priority than asking the effective questions and subsequent discussions due to the limited time for reviews. Facilitating effective questions and discussions after sharing potential true defects directly detected by reviewers may improve the review effectiveness. Furthermore, the results may imply that effective questions and discussions trigger the Phantom Inspector effect [58].

The case study suggests that NOQf and rNOD can be used as a metric to measure the effectiveness (quality) of reviews because metrics $p_3$ and $p_4$ affected $Q$. The metric can help a project manager (review leader) identify an insufficient review. This has two benefits. First, it reveals that the project manager should plan additional reviews with expert reviewers. Second, it highlights the need for more resources for subsequent testing.

*5.2. Implications for Practitioners*

The number of effective questions categorized as false-positive defects can be used as a metric to measure the software quality required by process models. The reviewers commented that the proposed metric can meet the requirements in process areas QPM.SP.1.4 in the CMMI and MAN.6.BP4 in the Automotive SPICE. The proposed method has a high usability from two perspectives. First, it can be used in various types of reviews, including code reviews with support tools. Second, it can objectively determine whether a question is effective, and the number of effective questions categorized as false-positive defects can be measured easily. Moreover, the proposed method is efficient even for cost-sensitive software development because categorizing effective questions and measuring the number of them can be performed in a short time.

In an iterative development process including agile development [59,60], the proposed method can predict the product quality in each iteration. Although design reviews might not be implicitly performed in some iterative development processes, the essence of the proposed method can be applied for architectural and implementation discussions or comments in code reviews.

*5.3. Threats to Validity*

In the case study, the criterion for distinguishing effective questions from other questions may be biased. However, the reviewers in the case study selected effective questions based on whether or not the question identified a concern. Furthermore, after the reviewers selected the effective questions, an assessor in the quality assurance department verified that each effective question identified a concern.

In the case study, the variance of difficulties for projects may affect $Q$. However, the standard development process should mitigate such variance. For projects with technical challenges such as deploying novel technologies, prior development and verification were conducted. For projects whose members did not have sufficient domain knowledge on the product, additional developers and reviewers with sufficient domain knowledge were invited to the reviews.

Identifying effective questions is potentially difficult. However, in this case study, the reviewers asked questions and categorized effective questions. Furthermore, identifying effective questions has previously been reported. One study [15] showed that some of the review comments could be categorized as questions, and about half of the approximately 470 questions helped reviewers detect defects.

## 6. Conclusions

This paper proposed a review metric measuring the number of effective questions, which identifies concerns about potential defects. Effective questions and subsequent discussions lead to defect detections if concerns identified by the effective questions and discussions are applicable to review materials, whereas concerns that are not applicable are discarded or recorded as false-positive defects. We performed a literature review to investigate whether previous studies referred to such effective questions and concerns. The results of the literature review showed that no article referred to questions and concerns that were categorized into true or false-positive defects. Additionally, no literature categorized true defects into defects directly detected and shared by a reviewer or those found by concerns identified by effective questions and subsequent discussions.

We conducted a case study to investigate the effectiveness of the metric. The case study measured the number of questions, which ensures that the authors and reviewers do not overlook defects in terms of the concerns identified by the questions. The case study evaluated the impact of the number of effective questions on the number of defects in subsequent testing by multiple regression analysis. The independent variables were the proportion of reused lines of code, proportion of true defects detected in reviews to the product size, proportion of effective questions categorized as false-positive defects in reviews to the product size, and proportion of effective questions categorized as false-positive defects to the sum of true defects and effective questions categorized as false-positive defects. The dependent variable was the proportion of the defects detected in testing to the product size. The evaluation used metrics collected in 25 projects in a company. As the proportion of the number of effective questions categorized as false-positive defects to the sum of the number of true defects and effective questions categorized as false-positive defects (ranging from 0.00 to 0.29) increased, the proportion of the number of defects detected in testing to the product size decreased (ranging from 0.54 to 6.50) ($b = -9.68$, $p = 0.02$). Additionally, as the proportion of the number of effective questions categorized as false-positive defects to the product size (ranging from 0.00 to 7.90) slightly increased, the proportion of the number of defects detected in testing to the product size increased ($b = 0.42$, $p = 0.01$).

Future works include (semi-)automatic categorization. Sentiment analysis is widely used in natural language processing research [61,62]. Recent studies have shown that sentiment analysis can categorize review comments from certain perspectives. For example, the sentiment of a comment (i.e., whether or not a comment is formulated in a positive or negative tone) may relate to comment usefulness [10], a model algorithm founded to identify review comments expressing negative sentiments [63], and the emotionality of the comment reflecting conventional metrics such as typing duration and typing speed [64]. Applying these studies to review comments may categorize effective questions that identify concerns.

**Author Contributions:** Conceptualization, M.W. and S.M.; Data curation, M.W.; Case study analysis, M.W.; Methodology, M.W. and S.M.; Supervision, S.M.; Visualization, M.W.; Writing—original draft, M.W.; Writing—review & editing, M.W. and S.M. All authors have read and agreed to the published version of the manuscript.

**Funding:** This research received no external funding.

**Informed Consent Statement:** Not applicable.

**Conflicts of Interest:** The authors declare no conflict of interest.

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
