# Peer review of "A Metric for Questions and Discussions Identifying Concerns in Software Reviews"

_2674-113X, doi:10.3390/software1030016_

Round 1
Reviewer 1 Report
The paper presents a metric that correlates effective questions to potential defects in software review. Although this is not a recent research topic, it is still important in software development.
In general, the proposal is well defined and validated in a case study using 25 projects. However, it seems to be an old research that should be updated for the current development scenario.
Some topics that should be improved:
- I recommend an additional background section with key concepts related to software review;
- Section 2.3 presents a literature review to identify works that describe concerns raised by effective questions and categorization of true and false positive defects. It would be good to add a paragraph positioning your proposal in relation to the other works, i.e. identifying the strengths of your proposal in relation to the others;
-Figure 4: The example says “the argument is designed for compatibility and is not used intentionally”. It could be difficult to identify in a review the “intention” of some design decisions;
-According to the paper, the proposal was evaluated in projects developed with a waterfall process model. However, currently the incremental (agile) process model has been widely adopted by organizations. It would be important, therefore, to position the applicability of the proposal in this development process model;
- Did you measure the time spent using your proposal during the case studies? All 25 projects are embedded systems, including safety-critical system software. In other domains, is it worth it? I think you could have a discussion, maybe in the threats to validity section, and talk about generalizing to other domains;
-Why do the authors take so long to present these results? According to the article, the study was validated with projects from 2010 to 2016.
Reviewer 2 Report
1. The concept of effective questions is valuable, which connects the concerns in the requirements with the possible system defects.
2. The concept definition of effective questions could be more formalized.
3. How to identify concerns and their locations about potential defects in questions can be explained in more detail.
4. The 25 projects in the case study could also be introduced in more detail.
5. The degree of automation of this method is slightly insufficient.
